

# Driving Mechanisms for Subsiding Shells in Simulations of Deep Moist Convection

Quinlan R. Mulhern[1], John M. Peters[1], and Jake P. Mulholland[2]

[1]Department of Meteorology and Atmospheric Science, The Pennsylvania State University, University Park, PA, USA
[2]Department of Atmospheric and Environmental Sciences, University at Albany, State University of New York, Albany, NY, USA

**Correspondence:** Quinlan R. Mulhern (qrm5008@psu.edu)

**Abstract.** Downdrafts play an essential role in the feedback between convective clouds and their surrounding environment, and they must be properly accounted for in cumulus parameterizations (CPs). The mechanisms for downdraft formation are often debated in past literature and inconsistently represented in CPs. To address this uncertainty, we investigate the ring of descent surrounding cloudy updrafts known as a subsiding shell, a leading contributor to downdraft mass flux. We analyze two LES of deep convection in the Amazonian dry and wet season, using composite soundings from the Green Ocean Amazon Campaign. The dry and wet season soundings differ in their middle tropospheric relative humidity (RH), which facilitates an assessment of the influence of RH on shell strength. Kinetic energy budgets along trajectories reveal that shells acquire their descent from evaporatively driven negative effective buoyancy along cloud edge and downward oriented dynamic pressure accelerations associated with the toroidal circulations of updraft thermals. Consistent with observations, shell downdrafts were strongest in the dry season simulation. Contrary to hypotheses which attributed this difference to greater evaporative cooling, we find that dry season shell downdrafts associated with deep convection were stronger because of larger dynamic pressure accelerations in the dry season. However, when investigating cumulus congestus clouds, negative effective buoyancy accelerations become increasingly important relative to pressure accelerations. The stronger accelerations in deep convective shells were attributed to stronger dry season updrafts, and consequently more intense toroidal circulations within thermals. Our results provide a foundation of understanding for future improvement of downdraft representation in CPs.

## 1 Introduction

Since the turn of the century, global climate models (GCMs) have been used to further comprehend the future state of Earth's climate. However, with the consideration of computational expense, GCMs are run at coarse resolutions and cannot explicitly resolve important microscale and mesoscale features such as deep convection. Therefore, models rely heavily on parameterization schemes that account for sub-grid scale processes. One such scheme used in many climate models is a cumulus parameterization.

Cumulus parameterizations typically use a simplified model for updrafts and downdrafts to represent their sub-grid scale fluxes. The sub-grid scale vertical flux of water vapor can be organized into terms that consider both downdraft and updraft detrainment, as well as large scale subsidence (Bretherton and Smolarkiewicz, 1989). The detrainment of liquid water from





a cloud into its environment is largely dependent on the vertical mass flux of updrafts associated with the clouds, and such
detrainment allows for moistening of the ambient environment directly (Savre, 2021). Additionally, mass, momentum, and
energy are transported downward via environmental subsidence away from convective clouds, which acts to warm and dry
the surrounding environment. Parameterization schemes typically ensure that the sum of upward and downward fluxes equals
the grid scale mass flux (Raymond, 1993). Though this handling of subsidence and downward mass transport appears both
intuitive and practical, there are a few shortcomings. For instance, many parameterizations assume downdrafts displace mean
grid state properties downward (Tiedtke, 1989; Bechtold et al., 2008; Park et al., 2016). However, downdrafts in the near-cloud
environment may have substantially different properties than the mean grid state and may not follow the standard assumption
of grid-wide redistribution of mass, heat, and momentum.

A near-cloud downdraft feature that has recently received research attention is the region of negative vertical velocity ($w$)
immediately surrounding convective clouds. This subsidence area is commonly referred to as the subsiding shell and has been
observed in the vicinity of shallow cumulus via aircraft observations (Jonas, 1990; Rodts et al., 2003; Katzwinkel et al., 2014;
Mallaun et al., 2019). Subsiding shells are located at cloud edge, protect cumulus clouds from their environment by allowing
them to entrain pre-moistened air (Jonas, 1990; Rodts et al., 2003; Heus and Jonker, 2008; Dawe and Austin, 2011; Zhang
et al., 2016; Hannah, 2017; Savre, 2021), and additionally serve as the primary contributor to downward heat and moisture
transport above the boundary layer rather than convective compensating subsidence (Glenn and Krueger, 2014; Park et al.,
2016).

There remains a debate concerning the mechanisms that drive downward accelerations in subsiding shells. Observational
studies have found that shallow cumulus subsiding shells are primarily driven by negative thermal buoyancy $B$ that arises
from the evaporation of cloud droplets at cloud edge (Rodts et al., 2003; Siebert et al., 2006; Katzwinkel et al., 2014; Mallaun
et al., 2019). In the presence of stronger cumulus updrafts, subsidence within the shell may further strengthen cloud edge
turbulent mixing, increasing evaporative cooling and shell thickness in a positive feedback loop (Siebert et al., 2006; Nair
et al., 2020). Estimates of entrainment in large eddy simulations (LESs) that assume clouds entrain unmodified air from the
far field environment often underestimate the fractional entrainment rates of clouds (Romps, 2010; Dawe and Austin, 2011;
Hannah, 2017) because these estimates neglect the moisture of shell air. Heus and Jonker (2008) confirmed via LES that
evaporative cooling was the sole mechanism for shallow, non-precipitating cumulus subsiding shell formation and acceleration.
Additionally, subsidence from downward $B$-driven accelerations was counteracted by upward vertical perturbation pressure
gradient accelerations ($VPG$) within the shell (Heus and Jonker, 2008). Rodts et al. (2003) corroborated the predominately
evaporation-driven nature of shells, noting that shells driven by pressure perturbations would contain a lower water vapor
mixing ratio ($q_v$) than the surrounding environment since mass would be transported downward from cloud top, which is
inconsistent with the aforementioned observational and modeling studies.

Additional mechanisms for subsiding shell formation have been explored as well. While finding some evidence for negative
$B$ as a driver in shell subsidence, Park et al. (2017) also noted that the spatial structure of shallow cumulus subsiding shells is
consistent with the simple assumption that updraft thermals yield a vortex feature driven by horizontal $B$ gradients whose outer
branch drives the shell downward along cloud edge (Sherwood et al., 2013; Romps and Charn, 2015; Park et al., 2016). Jonas





(1990) observed that shells encapsulating shallow maritime cumulus were formed via cloud-top pressure perturbations and the spherical vortex feature that arises from them, which is similar to Hill's vortex (Hill and Henrici, 1894), and were strengthened by evaporative cooling at cloud edge. Hence, it also appears possible that shells are sometimes driven by a $VPG$, and there is a lack of reconciliation between evaporation-driven and $VPG$-driven paradigms.

Generally, observational and LES studies focusing on shell formation from $B$ reversal/evaporative cooling or toroidal circu-
lations from cloud-top pressure perturbations have been confined to non-precipitating shallow cumulus clouds. However, aside from the work of Sherwood et al. (2013), Glenn and Krueger (2014), and Savre (2021), properties and formation mechanisms of subsiding shells surrounding tropical deep convective clouds are less commonly studied. Glenn and Krueger (2014) determined that compensatory downward mass flux associated with deep convection subsiding shells is 5% – 10% of upward flux at any given level, highlighting their importance for mass transport. Such deep convective updrafts are susceptible to the effects
of dynamic and turbulent entrainment (Morrison, 2016), which would yield strong evaporative cooling at cloud edge and shell formation. Additionally, thermal-like deep convective clouds with stronger updrafts generate cloud-top pressure perturbations that are greater in magnitude than that of shallow cumulus with significantly weaker updrafts (Parish and Leon, 2013; Morrison and Peters, 2018), increasing the potential importance of dynamic forcing of shells. Using LES, Savre (2021) determined that shells associated with deep moist convection were created by both mechanical, $VPG$-driven forcing near cloud top and
by evaporative cooling away from cloud top. However, dynamic pressure forcing near cloud top appeared critical in Savre (2021) for explaining the magnitude of negative $w$ in the shells. Given the variety of mechanisms driving shells in past literature, it is possible that the contributing mechanisms driving shell development in shallow cumulus and deep convection are environmentally dependent.

One natural laboratory for convection studies is the Amazon Rainforest in South America – a region where deep convection
occurs relatively often (Giangrande et al., 2023). The Green Ocean Amazon (GOAMAZON) field campaign, performed in 2014 and 2015, supplied ample data for the analysis of deep convective clouds and their associated downdraft structures. Giangrande et al. (2023) determined that there were two distinct convective seasons that affected downdraft characteristics in the Amazon: a dry and wet season, both characterized by nearly identical temperature profiles but varying in free tropospheric relative humidity (RH) (Giangrande et al., 2020, 2023). The dry season, which persists from June through September, is
characterized by larger most-unstable convective available potential energy (MUCAPE), while the wet season (December – April) has reduced MUCAPE and marginally less vertical wind shear (Giangrande et al., 2020, 2023). Vertically oriented radar wind profilers observed a higher incidence of strong downdrafts in the dry season than in the wet season. There are a variety of potential explanations for this difference. One hypothesis offered in their study was:

– downdrafts occurring in the vicinity of updrafts are stronger in the dry season because the drier air in the free troposphere
in the dry season led to greater evaporative cooling along cloud edge (Giangrande et al., 2023).

This hypothesis implies a dominant role of $B$ in modulating downdraft strength, and that environmental variations in downdraft strength are primarily modulated by thermodynamic differences aloft, such as whether the free troposphere is moist or dry.





However, Giangrande et al. (2023) also found some differences in updraft behavior including stronger updrafts at low-levels in the dry season. Hence, an equally plausible hypothesis is:

– differences in updraft strength contributed to commensurate differences in downdraft strength via the downward-oriented $VPG$ within rising cloud thermals.

We intend to leverage these environmental, updraft, and downdraft observations sampled in GOAMAZON to address the aforementioned hypotheses, providing reconciliation of past ambiguity about the mechanisms that drive subsiding shells in deep convection. Two LESs and a Lagrangian parcel trajectory analysis are used to further assess the accelerations driving
shell formation and maintenance. The next section provides a description of the LES setup and trajectory analysis used for shell investigation. Results of shell properties, the role of environmental RH in shell formation, and the comparison between deep and congestus shells will follow. Conclusions from this work and future research will be established in the final section.

## 2   Methodology

Two idealized LESs of deep moist convection were run using Cloud Model 1 version 21.0 (CM1; Bryan and Fritsch, 2002).
Both LES were run with an isotropic 100 m grid spacing over a 200 km$^2$ domain, with a depth of 20 km. The simulations differed from each other only with respect to their input atmospheric profiles. One simulation was initialized with the GOA-MAZON wet season composite sounding (referred to the "wet simulation"). Similarly, the second simulation was run using the dry season profile (referred to as the "dry simulation"). Each sounding included in the composited sounding was taken at 12 UTC before a convective event. Though both seasons have similar temperature profiles, we set them equal to each other (both
having that of the dry season; Fig. 1a,b) for simplicity to isolate the influences of RH on downdrafts (Fig. 2). The moisture profile of the boundary layer in both simulations is similar, but the RH spread between the two simulations increases above 2 km. In much of the free troposphere, the wet simulation RH magnitudes are 20% – 40% greater than the dry simulation at any given height. RH magnitudes become comparable above 11 km, but most deep convective clouds that are investigated in this study do not reach this height. The wind profiles of the simulations also differ with slightly stronger vertical wind shear present
in the dry season than in the wet season.





**Figure 1.** Skew-T Log-P diagrams of initial GOAMAZON (a) wet and (b) dry season composited soundings used for LESs. Panels (c) and (d) are simulated soundings 10 h into the simulation for the wet and dry season, respectively. The red and green lines are environmental temperature and dewpoint temperature, respectively. Black curves represent the density temperature of an undiluted lifted surface parcel. Wind barbs are in units of kts and are plotted on the right side of each panel. Surface-based CAPE is listed in the legends, calculated using the adiabatic lapse rate formulas from Peters et al. (2022b) and Peters (2024).



Sub-grid scale turbulence was parameterized using the turbulent kinetic energy scheme (e.g., Deardorff, 1980) and Morrison double-moment microphysics (Morrison et al., 2009) were used in both simulations. Lateral boundary conditions were periodic and both vertical boundaries were free slip. Simulations were spun-up with random potential temperature perturbations of $\pm$ 0.25 K sampled from a uniform distribution, along with a constant surface potential temperature flux of 0.06 K m s$^{-1}$ (a typical late morning value for Amazon heat flux; e.g., Malhi et al. 2002; da Rocha et al. 2004), and a constant water vapor mixing ratio flux of $2 \times 10^{-4}$ kg kg$^{-1}$ m s$^{-1}$ (chosen via trial-and-error to prevent over-drying of the boundary layer). All other model settings were left at their default values available in the CM1 version 21.0 source code.

Because detrainment from simulated convection led to large scale moistening of the ambient environment, nudging of the domain-averaged $q_v$ toward the initial moisture profile for each simulation was applied above 4 km with a time scale of 6 h over the course of the simulation to isolate the RH impacts on convection. This allowed for the environments in the simulations to retain their distinctly different initial free-tropospheric moisture profiles (Fig. 2a), characteristic of differences between the wet and dry seasons. RH profiles after a 10 h spin-up period are shown in Fig. 2b. Nudging was only applied above 4 km to allow for surface fluxes to grow a well-mixed boundary layer unimpeded.

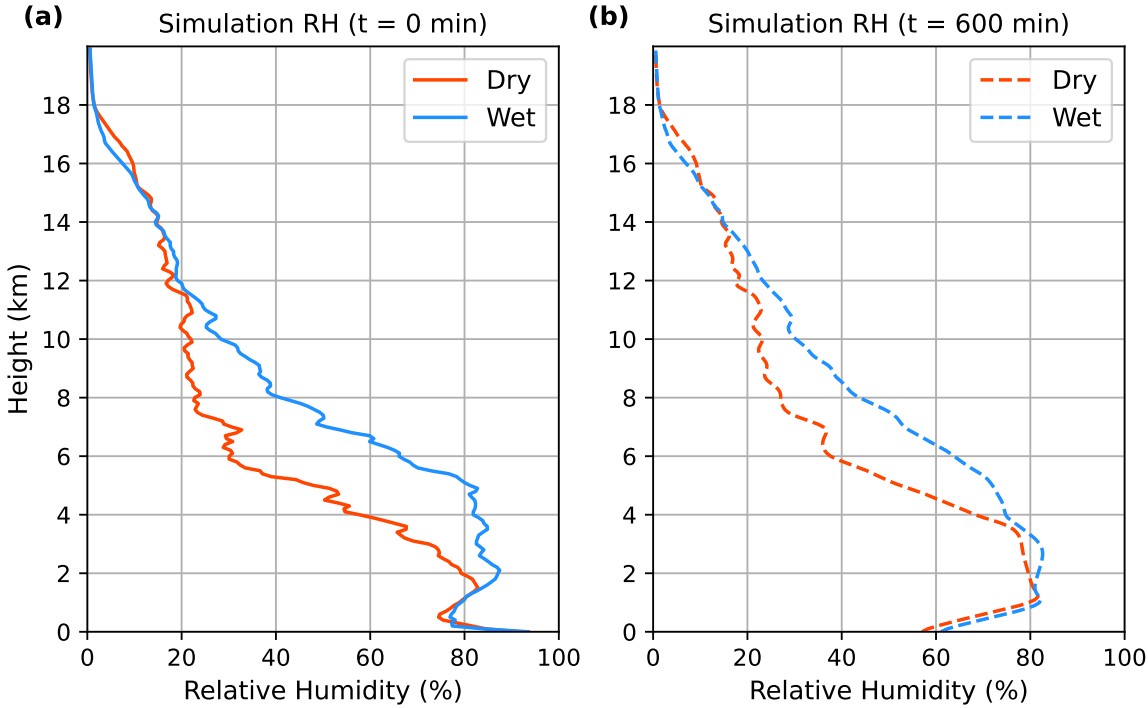

**Figure 2.** (a) Initial and (b) hour 10 profiles of environmental RH (%, x-axis) as a function of height (km, y-axis) for the dry (red) and wet (blue) simulation.





Maximum $w$ became steady in both simulations by 10 h, suggesting that the cloud population in both simulations had
achieved an approximate steady state. At this point, model output frequency increased to 30 s for the following hour. This
allowed for direct comparison between parcel properties and model properties without the need for temporal interpolation.
Additionally, all analysis of subsiding shell properties was performed during this hour.

Subsiding shells were identified following the methodology of Savre (2021) by first defining clouds as regions with total
condensate $(q_c + q_i)$ greater than 0.01 g kg$^{-1}$, where $q_c$ and $q_i$ are cloud water and ice mixing ratios, respectively. Cloud edge
was then defined where cloudy points were immediately adjacent to non-cloudy points. Following cloud edge identification,
an algorithm was applied to step outward from cloud edge and add all continuous grid points in the horizontal and vertical
where $w < 0$ m s$^{-1}$. There was no $B$ criterion included in the search to avoid making an a-priori assumption that subsiding
shell downdrafts are solely driven by negative $B$.

Parcel trajectories were tracked through subsiding shells using the built-in Lagrangian parcel trajectory analysis included in
the CM1 software package. At the initial time step, 5,494,536 parcels were distributed every 100 m from 1.5 km to 15 km in
the vertical. Parcel location, $w$, $B$, and $VPG$ accelerations were output every 30 s for each parcel during the analysis period.
Buoyancy in CM1 is computed via the following formula:

$$B = g\left(\frac{\theta - \theta_0}{\theta_0}\right) + g\left(\frac{R_v}{R_d} - 1\right)(q_v - q_{v,0}) - gq_{con}, \tag{1}$$

where g is the gravitational constant, $\theta$ is potential temperature, $R_v$ and $R_d$ are the specific gas constants for water vapor and
dry air respectively, $q_{con}$ is the total condensate mixing ratio, and 0 subscripts denote the height-only dependent initial model
profile (i.e., the soundings represented in Fig. 1a,b).

Because of the hydrostatic responses to the input of surface fluxes and adjustments of the large-scale atmosphere to convec-
tion, horizontal mean layers of the model domain attain large positive and negative $B$ after several hours of simulation. These
layers are hydrostatically balanced, and do not contribute meaningfully to the accelerations occurring within subsiding shells.
To make for cleaner physical interpretation of the distinct physical processes contributing to $B$ accelerations, the horizontal
mean of $B$ on a given level at each time was subtracted from the value of $B$ on that level at a given grid point, and added to the
$VPG$ at that grid point. This adjustment removes the horizontally-averaged hydrostatically balanced $B$, isolating the local $B$
relevant to convective processes.

Downdraft parcels that passed through subsiding shells were analyzed. Initially, downdraft parcels were defined as those
that met three criteria. Parcels must (a) begin their downdraft at a height of 5 km or greater, (b) reach a minimum $w$ of -1.5 m
s$^{-1}$ or stronger, and (c) undergo a permanent vertical displacement of at least 500 m in one downdraft swath. The height and
$w$ criterion were implemented to ensure parcels were associated with deep convective downdrafts and not within a large-scale
subsidence or gravity wave regime. Following this filtering, only parcels that passed through subsiding shell points for 90% of
their downdraft life (period where $w < 0$) were analyzed.

The prognostic pressure variable in CM1 is $\pi = \left(\frac{p}{p_{ref}}\right)^{\frac{R_d}{c_p}}$, where $p$ is pressure, $p_{ref} = 1000$ hPa is a reference pressure,
$R_d$ is the specific gas constant for dry air, and $c_p$ is the specific heat capacity at constant pressure. To attribute vertical pres-



sure accelerations to distinct physical processes, we decompose $\pi$ into buoyant ($\pi'_B$) and dynamic ($\pi'_D$) contributions. These contributions are obtained from the following equations (e.g., Klemp and Rotunno, 1983):

$$\nabla \cdot (c_p \theta_{0,\rho} \nabla \pi'_B) = \frac{\partial(\rho_0 B)}{\partial z}, \tag{2}$$


$$\nabla \cdot (c_p \theta_{0,\rho} \nabla \pi'_D) = -\nabla \cdot (\rho_0 \boldsymbol{V} \cdot \nabla \boldsymbol{V}), \tag{3}$$

where $\theta_{0,\rho}$ and $\rho_0$ are the initial height-dependent density potential temperature and density, respectively, and $\boldsymbol{V}$ is the three-dimensional wind vector.

Further decomposition of $\pi'_D$ and the right-hand side of the above equation into linear ($\pi'_{DL}$) and nonlinear ($\pi'_{DN}$) terms

gives:

$$\nabla \cdot (c_p \theta_{0,\rho} \nabla \pi'_{DN}) = -\nabla \cdot (\rho_0 \boldsymbol{V}' \cdot \nabla \boldsymbol{V}'), \tag{4}$$

$$\nabla \cdot (c_p \theta_{0,\rho} \nabla \pi'_{DL}) = -\rho_0 \nabla w \cdot \frac{d\boldsymbol{V_0}}{dz}, \tag{5}$$

where $\boldsymbol{V_0}$ is the simulation's initial horizontal wind profile and $\boldsymbol{V}' \equiv \boldsymbol{V} - \boldsymbol{V_0}$. The direct calculation of $\pi'_B$ and $\pi'_{DL}$ was completed by using a Fourier transform method in the horizontal and a tri-diagonal solver in the vertical, following the method used

in CM1 (Bryan and Fritsch, 2002). $\pi'_{DN}$ was then calculated as a residual. Retrieval of $VPG$ accelerations via centered finite differences was completed following this decomposition. The decomposed $VPG$ accelerations that arise from the computation of perturbation pressure were mapped onto shell parcel trajectory locations using trilinear interpolation.

Physically, $\pi'_{DN}$ is dependent on the local vorticity and deformation magnitudes. It is regions of relative low $\pi'_{DN}$ that are associated with the toroidal circulation in thermal convection, and hence dynamically forced accelerations will show up in

the component of the $VPG$ driven by $\pi'_{DN}$. The linear dynamic pressure $\pi'_{DL}$ relates to the interaction of the updraft with the background environment shear profile. As we will see, the component of $VPG$ from $\pi'_{DL}$ is typically small relative to $\pi'_{DN}$ and $\pi'_B$ due to the relatively weak vertical shear in both the wet and dry season environments. $\pi'_B$ is inversely related to vertical gradients in $B$ itself, with high $\pi'_B$ occurring where $B$ decreases most rapidly with height. Hence, the $VPG$ from $\pi'_B$ is connected to $B$ within the updraft and subsiding shell regions, which are distinctly separate processes from the dynamical

accelerations driven by the toroidal circulations.

Because a large spectrum of cloud depths can be lumped into a "deep" convective regime, shell parcels and the shells themselves were investigated in two subgroups: a deep regime and a congestus regime. This was done to focus on mature and deepening convection, while removing shallow cumulus from the analysis. We defined congestus based on the criteria found in Jensen and Genio (2006), with congestus clouds having cloud top heights below 5 km and deep cloud tops exceeding 5 km.

Deep and congestus parcels were gathered by taking previously defined shell parcels and placing them into their respective group based on origin height.



To connect parcel accelerations to cloud features, vertical composites of shells and their parent updrafts were created for the hour of analysis. First, we identified cloudy updrafts as continuous regions of $w > 0$ m s$^{-1}$ and $q_c + q_i > 1 \times 10^{-5}$ kg kg$^{-1}$, with a maximum $w > 3$ m s$^{-1}$ and a depth of at least 1 km. We then found the centroid of this feature by averaging the position

of all grid points. A 3D box was then placed around the cloud with a horizontal width of 6 km and a height of 8 km. The shell minimum was then identified as the minimum $w$ within the box within 800 m of cloudy updraft edge. We then composited fields within a vertically integrated plane that intersected the shell minimum $w$ and updraft centroid. Lastly, all vectors from updraft maximum $w$ to shell minimum $w$ were rotated to be aligned on a common axis and all fields were normalized by fractional distance from updraft maximum $w$ to shell minimum $w$. $B$, $B_{eff}$, and $VPG$ accelerations were composited in the

same way, as well as $q_v$, $\pi'_B$, and $\pi'_{DN}$.

## 3 Results

### 3.1 Differences in Updraft Properties

A well-mixed layer formed adjacent to the lower boundary in both simulations and gradually deepened through the 10 hours preceding our analysis window (not shown). Cloud and updraft depths in both simulations also gradually deepened

during the 10-hour spin-up time period, eventually reaching quasi-steady heights in the 14-16 km range (Fig. 3). Updraft top heights peaked in the 12-16 km range in both simulations, which is broadly consistent with echo top heights observed from GOAMAZON (e.g., Fig. 1a,b in Giangrande et al., 2023). Also consistent with the updraft observations analyzed in Giangrande et al. (2023) (their Fig. 5a-c), time-averaged maximum $w$ was greater in the dry season simulation ($\sim 31$ m s$^{-1}$) than in the wet season simulation ($\sim 27$ m s$^{-1}$), and peak updraft intensities occurred lower in the atmosphere in the dry simulation than

in the wet season simulation (Fig. 3c), though updrafts above roughly 10 km were stronger in the wet season simulation.





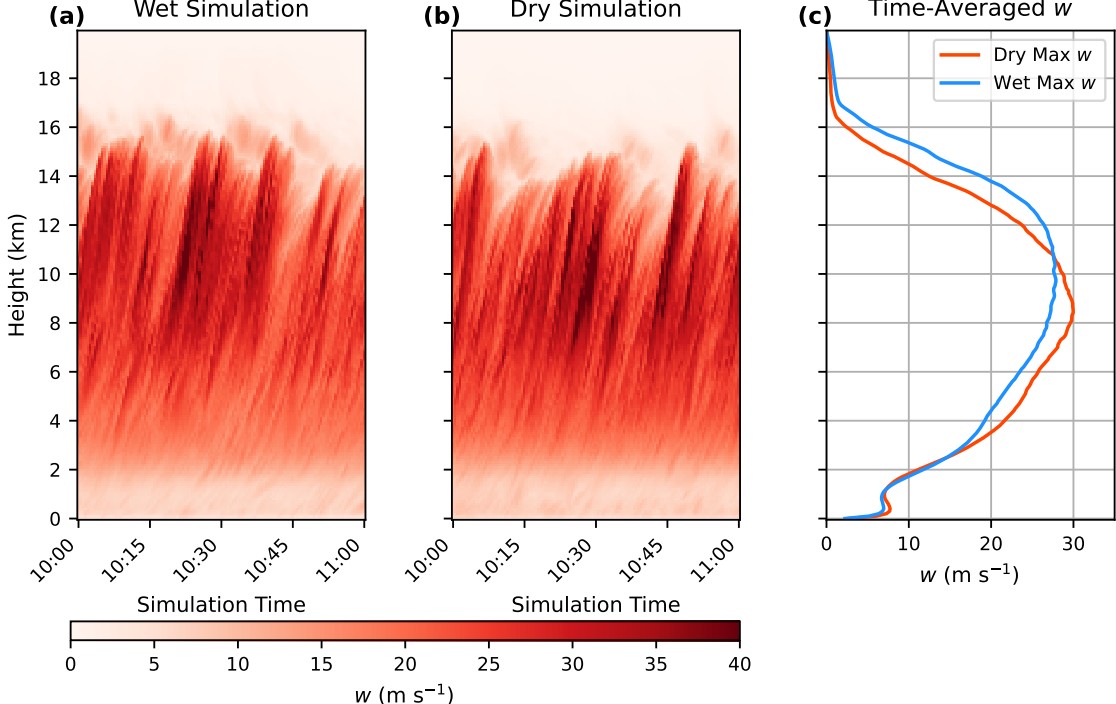

**Figure 3.** Hovmöller diagrams of horizontal domain maximum $w$ with height during the period of analysis for the (a) wet simulation and (b) dry simulation. Vertical profile of time-averaged $w$ during the analysis period for both simulations is exhibited in panel (c).

As will be shown later, these differences in updraft intensity and vertical distribution between the dry and wet season simulations were important contributors to commensurate differences in shell intensities. We demonstrate the physical reasons for these differences by examining the temperature and moisture, the $B$ of lifted surface parcels in the 10 h sounding, and by applying a simple updraft model. In this model, a surface parcel is lifted and diluted with a fractional entrainment rate $\varepsilon$

calculated from the following formula:

$$\varepsilon \equiv \frac{2k^2 L}{P_r R^2}, \tag{6}$$

where $k^2 = 0.18$ is the squared Von-Karman constant, $P_r = \frac{1}{3}$ is the turbulent Prandtl number, $L = 200$ m is a mixing length, and $R$ is the updraft radius. This formula was derived in Morrison (2017) based on an eddy diffusivity approximation for lateral mixing. We set $R$ to the lifting condensation level (LCL) height for a surface parcel calculated using the formula from Romps

(2017), consistent with past studies that have shown correlations between the LCL height and updraft radius (Mulholland et al., 2021). The thermodynamic properties and $B$ of the surface parcel were calculated using methods described in Peters et al. (2022b) and Peters (2024), which include hydrometeor loading and ice processes. Using this $B$ profile, $w$ was calculated using the updraft plume model from Peters et al. (2022a) (see their Appendix A).



Vertical profiles of $\Delta T_0 \equiv T_{0,wet} - T_{0,dry}$ (Fig. 4a), where the $T_0$ variables are the horizontal averages of $T$ shown in Fig.

1c,d, reveal warmer temperatures (i.e., positive $\Delta T_0$) at low-levels in the wet simulation than in the dry simulation at 10
h, particularly in the 2-7 km layer. This difference resulted in smaller undiluted adiabatic $B$ in the 2-7 km layer in the wet
simulation (blue dashed lines in Fig. 4b) than in the dry simulation (red dashed lines in Fig. 4b), which also translated to a
commensurate difference in diluted $B$ (solid lines in Fig. 4b). Using the profiles of diluted $B$, the updraft model (Fig. 4c)
yields profiles of wet season (solid blue) and dry season (solid red) $w$ that closely mimic the time averaged profiles from the

simulations (see Fig. 3c). The larger low-level diluted $B$ in the dry simulation partially explains the stronger low-level updraft
strengths in the dry simulation than in the wet simulation.

The differences in moisture between the wet and dry simulation also influenced the differences in $w$ among the simulations.
For instance, vertical profiles of $\Delta q_0 \equiv q_{0,wet} - q_{0,dry}$ reveal smaller surface and lower tropospheric moisture (green line is
positive in Fig. 4a) in the dry simulation. Because of this, the dry simulation developed a higher LCL height (1285 m) than the

wet simulation (1166 m), which contributed to wider updrafts and smaller $\varepsilon$ in the dry simulation, resulting in strengthened
updrafts relative to the wet simulation. We deduce this because neglecting this difference in $\varepsilon$ in the updraft model (left-most
dotted red line in Fig. 4c) results in weaker maximum $w$ in the dry profile than in the wet profile, which is inconsistent with
the simulations. On the other hand, a drier lower free troposphere enhanced dilution in the dry simulation relative to the wet,
which moderated $w$ in the dry simulation and resulted in stronger updrafts aloft in the wet simulation. We deduce this because

replacing the moisture profile in the dry profile with that of the wet and re-running the updraft model results in a dry season $w$
that is erroneously stronger than the wet season $w$ at all levels (right-most dotted red line in Fig. 4c).



**Figure 4.** Profiles of (a) $\Delta T_0$ (orange, K) and $\Delta q_0$ (green, g kg$^{-1}$) at 10 h ($\Delta$ indicates wet season minus dry season), (b) undiluted $B$ (dashed lines) and diluted $B$ (solid lines) for parcels lifted from the surface using the 10 h horizontal mean wet season (blue) and dry season (red) profiles, and (c) plume model $w$ (m s$^{-1}$) predicted from the 10 h horizontal mean profiles (solid lines) for the wet season (blue) and dry season (red) simulations. Dashed red lines depict plume model $w$ with the $\varepsilon$ from the wet season applied to the dry season parcels (left dashed line) and the $q_0$ profile from the wet season applied to the dry season profile (right dashed line).





## 3.2 Driving Mechanisms for Subsiding Shells

Utilizing parcel trajectories, subsiding shell properties were assessed in bulk. Dry and wet simulation shell parcel $w$ are displayed in Fig. 5. To avoid performing a temporal interpolation of trajectories, the first downdraft point for each parcel is

the first time step in which $w < 0$ m s$^{-1}$. As a result, trajectory $w$ does not begin at 0 m s$^{-1}$, but rather at some value below it. Immediately, it is evident that dry shell downdrafts are stronger than those in the wet simulation, consistent with the RWP observations in Giangrande et al. (2023). Mean wet simulation shell parcel $w$ reached a mean minimum of -2.78 m s$^{-1}$, whereas dry shell parcels achieved a mean minimum $w$ of -3.44 m s$^{-1}$. This equates to dry simulation downdrafts being 23.5% stronger than their wet simulation counterparts. Statistical significance in the difference of the means was determined using a Welch's

t-test with a significance level ($\alpha$) of 0.05. Mean differences were statistically significant through the entire downdraft period, with dry simulation downdrafts on average being 26.5% stronger than those in the wet simulation.

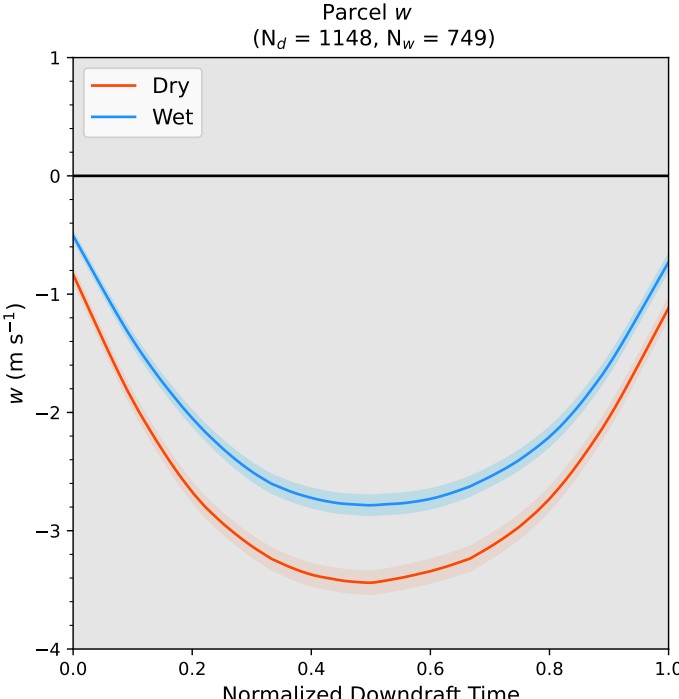

**Figure 5.** Mean subsiding shell parcel $w$ through normalized downdraft time for the dry (red) and wet (blue) simulations. 95% confidence intervals are shaded for each simulation and parcel sample size is noted above each panel. Grey shading indicates areas where simulation differences are statistically significant with $\alpha = 0.05$. $N_d$ and $N_w$ denote the sample size of dry and wet simulation parcels, respectively.

Mean accelerations acting on subsiding shell parcels are shown in Fig. 6. With regard to total accelerations, it is evident in both simulations that parcels that pass through shells are both negatively buoyant and are initially accelerated downward by





$VPG$ accelerations. Though not shown, parcels experienced minima in downward accelerations from both total $VPG$ and $B$

a few seconds to minutes before achieving a negative $w$.

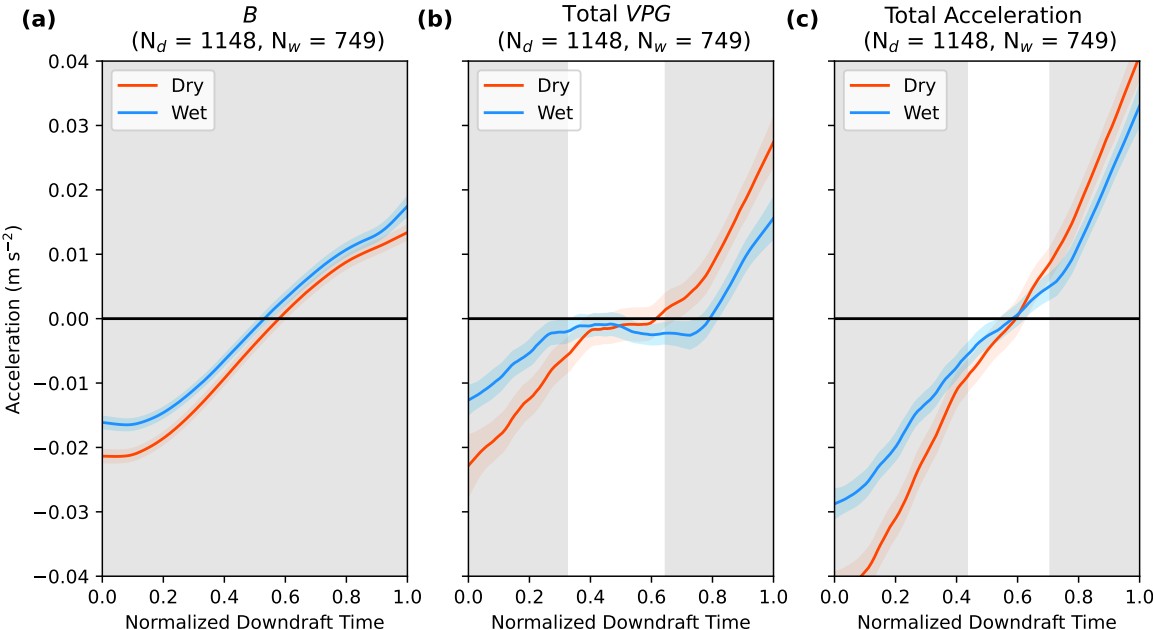

**Figure 6.** Mean subsiding shell parcel (a) $B$, (b) $VPG$, and (c) total acceleration through normalized downdraft time for the dry (red) and wet (blue) simulations. 95% confidence intervals are shaded for each simulation and parcel sample size is noted above each panel. Grey shading indicates areas where simulation mean differences are statistically significant with $\alpha = 0.05$.

In Fig. 6a, subsiding shell parcels in the dry simulation are initially more negatively buoyant with a minimum average $B$ of -0.021 m s$^{-2}$. $B$ then increases through $\approx$55% of the downdraft time, at which point $B$ sign reversal occurs and $B$ becomes positive due to parcels overshooting their level of neutral $B$. Wet simulation mean $B$, which is most negative at shell downdraft initialization with a value of -0.016 m s$^{-2}$, shares a similar increasing trend to that of the dry simulation. Like $B$,

the $VPG$ (Fig. 6b) is initially negative and decreases in magnitude as parcels descend. With an initial $VPG$ of -0.023 m s$^{-2}$, dry simulation $VPG$ acceleration becomes positive $\approx$60% of the way through the downdraft before reaching a maximum of 0.027 m s$^{-2}$ at downdraft end. However, unlike $B$ which increases continuously, dry simulation $VPG$ accelerations plateau halfway through the downdraft. The wet simulation $VPG$ undergoes the same behavior, albeit with both smaller downward accelerations initially, and smaller upward accelerations later on. Statistical significance of $B$ and $VPG$ acceleration behaviors

was assessed as well. $B$ differences were significant through the entirety of the downdraft, while $VPG$ differences showed significance through the first and final $\approx$35% of the downdraft. With this, differences in total acceleration means are significant in both the cloud-top region where the shells initiate and the shell terminus.



The parcel buoyant ($VPG_B = -c_p\theta_{\rho,0}\frac{\partial\pi'_B}{\partial z}$), dynamic linear ($VPG_{DL} = -c_p\theta_{\rho,0}\frac{\partial\pi'_{DL}}{\partial z}$), and dynamic nonlinear ($VPG_{DN} = -c_p\theta_{\rho,0}\frac{\partial\pi'_{DN}}{\partial z}$) contributions to $VPG$ accelerations in both simulations are plotted in Fig. 7. Effective buoyancy ($B_{eff}$; i.e.,

the net thermodynamic acceleration, Davies-Jones, 2003), which is simply the sum of $B$ and $VPG_B$, is also plotted to show the effect that pressure perturbations have on modulating $B$.



**Figure 7.** Same as Fig. 6, but for (a) $VPG_{DN}$, (b) $VPG_B$, (c) $B_{eff}$, and (d) $VPG_{DL}$.

$VPG_{DN}$ has the largest magnitude of the three decomposed perturbation pressure accelerations. The temporal evolution of $VPG_{DN}$ closely mimics that of the total $VPG$. Initially $VPG_{DN}$ drives strong downward acceleration, but undergoes a sign




reversal and drives strong upward acceleration by the end of the downdraft lifetimes. The magnitude of the initial downward
$VPG_{DN}$ is considerably larger in the dry simulation than in the wet simulation, suggesting that this pressure contribution
largely explains the differing total $VPG$ magnitudes seen in Fig. 6c. Wet simulation $VPG_B$ begins as positive but quickly
becomes negative 20% into the downdraft, contributing downward acceleration thereafter. In contrast, the $VPG_B$ in the dry
simulation remains positive for $\approx$55% of the downdraft, becoming negative thereafter. Such $VPG_B$ accelerations partially
counteract the strong negative $B$ that arises from cloud edge evaporative cooling, while also opposing the positive $B$ that
decelerates downdraft parcels once they have descended below their levels of neutral $B$. Consequently, $B_{eff}$ shows a similar
increasing trend like $B$ itself, but magnitudes are reduced at the origin and termination points of the downdraft compared to $B$.
In fact, $B_{eff}$ magnitudes are reduced from $B$ to a greater extent in the dry simulation than in the wet because dry simulation
$VPG_B$ is positive for more of the downdraft. This counteracts the difference in $B$ between the simulations, leading to similar
$B_{eff}$ magnitudes. $VPG_{DL}$ is very small relative to the other $VPG$ contributions, which is likely a result of relatively weak
vertical wind shear in both simulations (Fig. 1). This pressure contribution does not contribute to the differing overall $VPG$
magnitudes seen in Fig. 6c.

Following Peters et al. (2019), we ascertain the net (i.e., integrated) contribution of each of these accelerations to downdrafts.
This is accomplished by first considering the frictionless vertical velocity equation:

$$\frac{dw}{dt} = B + VPG. \tag{7}$$

We may use the chain rule to re-write the left-hand-side term as $\frac{dw}{dt} = \frac{dw}{dz}\frac{dz}{dt} = w\frac{dw}{dz} = \frac{d}{dz}\frac{w^2}{2} = \frac{dKE}{dt}$, where $KE \equiv \frac{w^2}{2}$ is the
kinetic energy. Upon vertically integrating from the downdraft start height $z_i$ to any arbitrary height below the start height $z$,
we obtain:

$$\Delta KE_B(z) = \int\limits_{z^*=z_i}^{z^*=z} B dz^*, \tag{8}$$

$$\Delta KE_{VPG}(z) = \int\limits_{z^*=z_i}^{z^*=z} VPG dz^*, \tag{9}$$

$$\Delta KE_{tot}(z) = \Delta KE_B(z) + \Delta KE_{VPG}(z), \tag{10}$$

where $\Delta KE$ is the change in kinetic energy between the parcel's starting location $z_i$ and a given height below the starting
point $z$ and $z^*$ is a dummy variable of integration. We similarly compute $\Delta KE_{VPG,B}$, $\Delta KE_{VPG,DL}$, $\Delta KE_{VPG,DN}$, and
$\Delta KE_{B_{eff}}$. Note that despite downward velocities within a downdraft, the absolute change in $KE$ is positive since $w$ is
squared. Processes that accelerate the parcel downward will contribute positive $\Delta KE$, whereas parcels that accelerate the
parcel upward will contribute negative $\Delta KE$. With the assumption that $KE(z_i) = 0$ at the downdraft start, all $\Delta KE$ terms
can be expressed as $KE$.





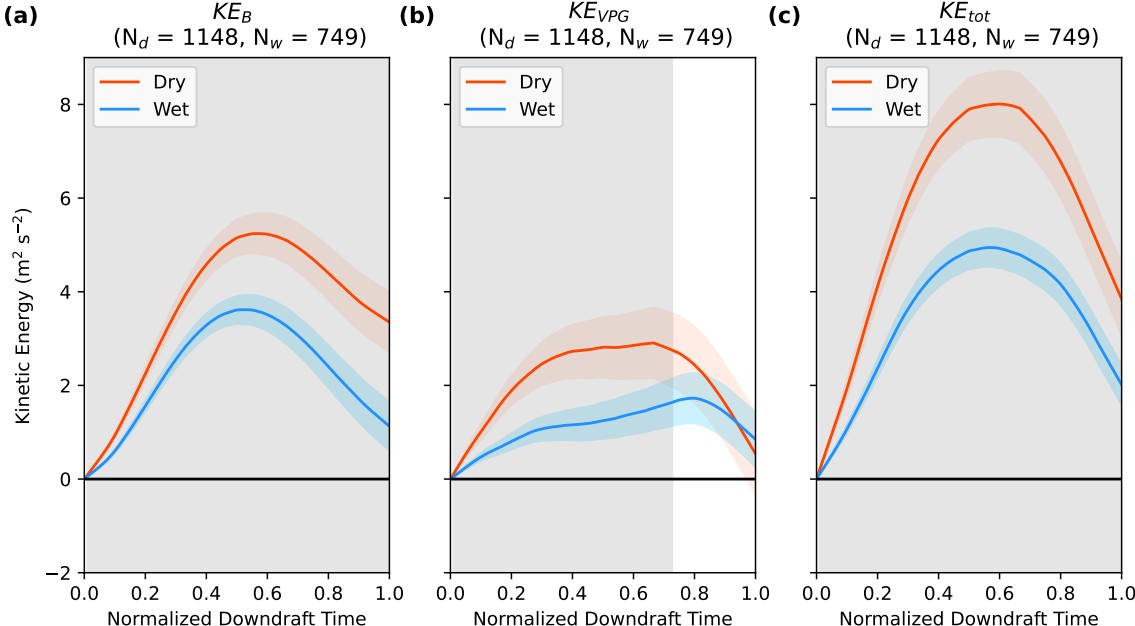

**Figure 8.** Same as Fig. 6, but for parcel $KE$.

Throughout the downdraft, $KE_{tot}$ is larger in the dry simulation than the wet (Fig. 8c) because downdraft speeds are more negative in the dry than in the wet simulation. Both $KE_B$ and $KE_{VPG}$ contribute meaningfully to total $KE$, though $KE_B$ (Fig. 8a) is roughly double that of $KE_{VPG}$ (Fig. 8b) in both simulations. Hence, downdrafts in our simulations are both buoyantly and dynamically driven, though $B$ is the predominant contributor. Both $KE_B$ and $KE_{VPG}$ are larger in the dry simulation by roughly equal magnitudes. This means that while $B$ is the predominant overall driver of shell downdrafts, the environmental differences in downdraft strength are roughly equally driven by differences in $B$ and differences in the $VPG$.

The results in Fig. 8 support both of the hypotheses proposed in the introduction that larger negative $B$ and larger downward dynamic pressure accelerations drive stronger downdrafts in the dry season than in the wet season. However, an examination of $KE_{B_{eff}}$ and $KE_{VPG,B}$ calls this conclusion to question. Specifically, the appreciably larger $KE_B$ in the dry season simulation is largely offset by smaller $KE_{VPG,B}$ in the dry season simulation, such that the difference in $KE_{B_{eff}}$ between the wet and dry season simulations is small (Fig. 9c). This implies that while downdrafts are more negatively buoyant in the dry season, the atmosphere's thermodynamic response to these more negatively buoyant downdrafts induces an upward pressure gradient acceleration that nearly entirely offsets this larger negative $B$. Hence, the differences in downdraft strength between the wet season and dry season are predominately driven by larger $KE_{VPG,DN}$ in the dry season than in the wet season (Fig. 9a). In other words, there is stronger dynamic forcing for downdrafts in the dry season. Furthermore, $KE_{B_{eff}}$ is now comparable in magnitude to $KE_{VPG,DN}$, implying that shells are roughly equally thermodynamically and dynamically forced.







**Figure 9.** Same as Fig. 8, but for (a) $KE_{VPG,DN}$, (b) $KE_{VPG,B}$, (c) $KE_{B_{eff}}$, and (d) $KE_{VPG,DL}$.





## 3.3 Congestus Versus Deep Shells

Exploration of shell origin and termination heights prompted a comparison between shells associated with shallow/congestus
cumulus and deep convection (Fig. 10).





**Figure 10.** Probability density histograms for wet and dry simulation shell parcel (a) origin and (b) termination heights. Panel (c) shows displacement of the shell parcels for both simulations.



A bimodal distribution appeared in both the origin and termination heights, with a peak in origin heights centered about 9000 m and 8000 m in the dry and wet season, respectively, but also at around 3000 m in both simulations. Termination height peaks in the distribution are shifted downward by about 500 – 1000 m, roughly matching the displacement magnitudes. We speculate that the bimodal distribution in shell origin/termination heights is associated with the congestus and deep regimes of

cumulus. Given the large difference in the heights, it is possible that shell properties for congestus and deep convection may be different, especially since RH values are higher in the lower troposphere and the updrafts of congestus clouds are weaker. For this comparison, we assume cumulii that have a cloud top greater than 5 km are considered deep (Jensen and Genio, 2006), whereas any cloud below this height and above 2 km is categorized as congestus. Mean $w$ of parcels passing through both deep and congestus shells is shown in Fig. 11.

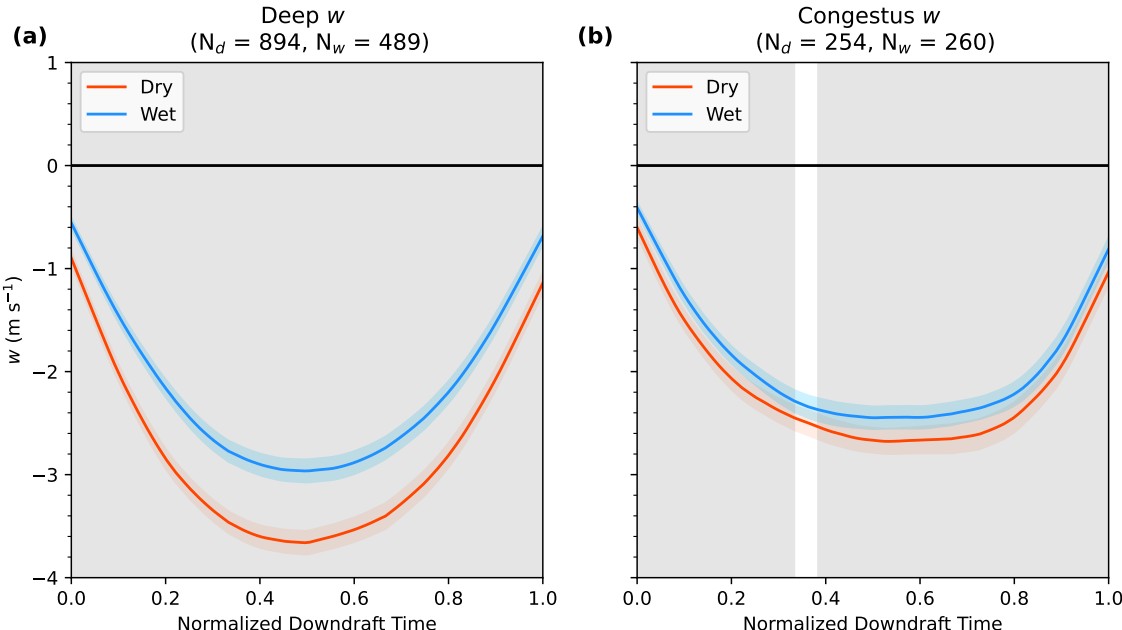

**Figure 11.** Same as Fig. 5, but for (a) deep and (b) congestus regimes of convection.

Unsurprisingly, the mean $w$ magnitude of deep parcels is greater in the deep convection shells than it is in the congestus shells. Deep shell parcels attain minimum $w$ of -3.66 m s$^{-1}$ and -2.96 m s$^{-1}$ in the dry and wet simulations, respectively. In the congestus regime, dry parcel subsidence is 1 m s$^{-1}$ weaker than in the deep and wet parcel subsidence is only slightly reduced to -2.44 m s$^{-1}$.

With differing $w$ magnitudes between the two types of convection, we again perform an assessment of $KE$ contributions to

gauge the importance of thermodynamic and dynamic accelerations in shells. Figure 12 shows the comparison between deep and congestus contributions to $KE$ from $B_{eff}$ and the $VPG_{DN}$, as well as their sum.



**Figure 12.** Same as Fig. 8, but for (a, d) $KE_{B_{eff}}$, (b, e) $KE_{VPG,DN}$, and (c, f) their sum for (top) deep and (bottom) congestus regimes of convection.

As was the case with $w$, maximum $KE$ is greater in the deep shells versus the congestus shells (Fig. 12c,f). Additionally, in both convective regimes, dry simulation $KE$ is greater at all times compared to the wet simulation. Given the larger num-





ber of parcels embedded within deep shell downdrafts, the behavior of $KE$ in this regime is similar to that of all downdrafts

combined (Fig. 8). With this being the case, attention turns toward the congestus regime, where $KE$ contributions from $B_{eff}$ and $VPG_{DN}$ differ from their deep counterparts. Maximum total $KE$ in congestus shells is reduced from that in deep shells in both simulations, with dry and wet congestus shells on average having 28.8% and 24.8% less KE than deep shells, respectively. Despite $KE_{B_{eff}}$ more than doubling in the dry simulation, total $KE$ decreases. This is due to the overpowering effect of $KE_{VPG,DN}$, which decreases significantly and is largely negative for much of the downdraft in the congestus regime. Mag-

nitudes of wet simulation differences in individual accelerations are reduced, with both $KE_{B_{eff}}$ increasing and $KE_{VPG,DN}$ decreasing only minimally. With this, it is clear that $VPG$s are generally weaker (and potentially more detrimental) in congestus shells than they are in deep convection shells. Furthermore, $KE_{VPG,DN}$ is not statistically different between the dry season and wet season congestus shells, implying that the more intense downdrafts in dry congestus shells are primarily driven by these shells being more negatively buoyant in the dry season. This contrasts with deep convective shells, which were primarily

stronger in the dry simulation because of a stronger downward $VPG_{DN}$ in that simulation.

### 3.4 Physical processes contributing to accelerations

Visualization of the shell structures in both simulations and the two modes of convection was done via compositing of shells. The composites of subsiding shells allow us to physically interpret the processes driving the shells, and to understand how these processes differ between the two simulations. Figure 13 displays composited deep convective shells and their associated

accelerations. Focusing on $B$ in Fig. 13, a clear region of negative $B$ encapsulates a strongly positively buoyant updraft core. Negative $B$ exists both above the updraft and on the flank of the updraft where shells are present. Such negative $B$ may arise in two ways. The negative $B$ existing above the updraft arises from the adiabatic cooling of environmental air that is pushed upward by an ascending thermal. Here, negatively buoyant parcels are forced horizontally toward the flanks of the updraft and can begin to sink down the edges of the cloud in the shell. However, stronger negative $B$ exists where shell subsidence

is maximized. It is here that a strong positive perturbation of $q_v$ is present, likely owed to the evaporative cooling of cloud droplets at the edge of the updraft. The presence of this feature in the composite supports the evaporative cooling mechanism of formation of subsiding shells (e.g., Heus and Jonker, 2008). When comparing $B$ to $B_{eff}$, the region of downward acceleration that was located above the updraft is eliminated and only that which occurs in the shell remains. Because $B_{eff}$ accounts for the effects of $VPG_B$, and with a high $\pi'_B$ perturbation found in the upper portions of the updraft, almost all downward acceleration

from negative $B$ above the updraft is offset by upward $VPG_B$, leading to no downward acceleration in this region. Therefore, all subsidence only will occur on the flanks of the updraft in the shell where negative thermal $B$ arising from evaporative cooling dominates.

While a clear horizontal dipole in the $B$ and $B_{eff}$ structure exists between the shell and the updraft, a vertical dipole is present in the $VPG_{DN}$ field and is maximized where vertical gradients of $\pi'_{DN}$ are largest. The $\pi'_{DN}$ field is structured so that

a minimum in pressure is found embedded within the maximum horizontal gradient of $w$. This is also where the horizontal $B$ gradient is maximized, allowing the minimum in pressure to arise in the center of the toroidal circulation associated with rising cloud thermals. As seen in both the composites and parcel trajectory accelerations, this $VPG_{DN}$ dipole structure leads



to a downward acceleration above the shell core, and a positive acceleration below it. Additionally, the magnitude of $VPG_{DN}$ is much greater than that of $B$ or $B_{eff}$. This is especially the case where shell subsidence originates, about 500 m above
the region of maximum subsidence. It is here that parcels are essentially "sling-shotted" down the side of the cloud by the downward branch of the toroidal circulation, which acts as the driver of shell subsidence (e.g., Savre, 2021).

The dependence of downward shell accelerations on the toroidal low pressure regions explains why $VPG_{DN}$ is more intense in the dry simulation than in the wet simulation. That is, the magnitude of the toroidal low should scale with the magnitude of $w$ and the horizontal $B$ gradient within the cloud core. This connection between $w$ and $\pi'_{DN}$ is evident in the analytic solutions
of Hill and Henrici (1894), and is further shown in simulations of moist thermals in Morrison and Peters (2018) (see eq. 13 in that study). Hence, the dry simulations have stronger downdrafts than the wet because the dry simulations have stronger overall updrafts and consequently stronger downward accelerations driven by toroidal low pressure (Marion and Trapp, 2019).







**Figure 13.** Wet (left column) and dry (middle column) simulation composites of subsiding shell downdrafts for the deep regime of convection. The vertical axis is the height (in km) relative to the height of the shell minimum $w$, and the horizontal axis is the fractional distance between the maximum $w$ at the height of shell minimum $w$ and the shell minimum $w$ itself. Composites of $B$ (top), $B_{eff}$ (middle), and $VPG_{DN}$ (bottom) are filled, and simulation parcel trajectories for these accelerations are shown in the right column. $w$ is depicted in black contours, and $q_v'$ (top; in g kg$^{-1}$), $\pi_B'$ (middle) and $\pi_{DN}'$ (bottom) are contoured in grey. $\pi_B'$ and $\pi_{DN}'$ are scaled by $10^3$.







**Figure 14.** Same as Fig. 14, but for the congestus regime.





The structure of the $w$ field in the congestus regime composite (Fig. 14) is similar to that of the deep composite, aside from the congestus updrafts being weaker. The most notable differences between the deep and congestus composites is associated

with the shell features. Congestus composite negative $B$ and $B_{eff}$ wrap around the updraft in a bean-like shape, and negative values in these features extend further downward than they do in the deep convection composites. The trajectories also show this, with negative buoyant accelerations persisting in the downdraft period longer than in the deep shells. Therefore, it is likely that the congestus regime exhibits more of the evaporative cooling processes that occur at cloud edge, which is supported by slightly stronger $q_v'$ in the congestus shells than the deep shells. Though $B$ variations are present between the two regimes,

the most notable differences is embedded within the $VPG_{DN}$ fields. The composite structure of congestus $VPG_{DN}$ is nearly identical to that of deep convection, but magnitudes are reduced. This comes as no surprise as the strength of the $VPG_{DN}$ is proportional to horizontal $B$ gradients and updraft strength, and such gradients and updrafts are weaker in the congestus convection. The weakened negative $VPG_{DN}$ prevents shells from being as strong as those in the deep regime.







**Figure 15.** Schematic diagrams of parcel (colored circle) descent through a subsiding shell at cloud edge (contoured in black at cloud edge) when it is (a) above a thermal, (b) at its minimum $w$, and (c) below a thermal. Parcel color is blue when negatively buoyant and red when positively buoyant. Thermal circulations are denoted by bold black arrows around thermal-induced low pressure (depicted with L). Deep convective updrafts are denoted with red arrows passing through the center of the cloud. Accelerations (total, $B$, and $VPG$) are shown to the right of the cloud, and signs and relative magnitudes of the accelerations are implied with direction and length of the arrows.





To summarize (e.g., Fig. 15) parcels near cloud top are first accelerated downward by the $VPG_{DN}$ induced by the toroidal circulation and by negative $B$ driven by the evaporation of condensates. Because this circulation is proportional to updraft strength and horizontal $B$ gradients, the $VPG_{DN}$ is stronger in deep convection versus shallow convection, and in stronger dry season simulations than in the wet season simulations. As soon as parcels within the shell pass below the height of minimum $\pi'_{DN}$ in the toroidal circulation, they experience an upward $VPG_{DN}$ acceleration and only $B_{eff}$ and momentum can maintain the subsidence. Parcels will also become more positively buoyant as they overshoot their levels of neutral $B$. Once parcels are well below the height of minimum $\pi'_{DN}$ and are positively buoyant, shell subsidence ceases. Based on both Fig. 10 and the composites, such processes yielded shells that were only about 500 m – 1500 m in depth, depending on the strength of the convection.

## 4   Conclusions

This study investigated the influence of environmental RH on subsiding shells in deep moist convection using LES and a Lagrangian parcel trajectory analysis. Two LESs were initialized with atmospheric profiles representing the convective regimes of the wet season and the dry season in the Amazon. The primary difference between these two atmospheric profiles was larger middle free-tropospheric RH in the wet season than in the dry season, which facilitates a systematic assessment of the influence of RH on subsiding shell behavior.

Shell parcels were typically displaced downward by roughly a kilometer, with maximum negative $w$ on the order of 2-4 m s$^{-1}$. Lagrangian parcel analysis reveals that the shells were driven by both negative effective buoyancy resulting from condensate evaporation and downward oriented dynamic pressure accelerations driven by the low pressure regions within the toroidal circulations of cloud thermals. Descent in downdrafts ceased once parcels overshot their level of neutral buoyancy and passed below the toroidal low, after which $B$ became positive and the pressure gradient acceleration became oriented upward.

Consistent with radar observations of the wet season and dry season in the Amazon (e.g., Giangrande et al., 2023), downdrafts in subsiding shells were significantly stronger in the dry season than in the wet season. While dry season shells intuitively experienced larger negative buoyancy than their wet season counterparts because of greater condensate evaporation in a drier lower free troposphere, this difference was largely counteracted by a larger upward oriented buoyancy pressure acceleration in the dry season. Hence, while large and downward in both simulations, the effective buoyancy was similar in dry season and wet seasons. In contrast, downward oriented dynamic pressure accelerations were significantly stronger in the dry season simulation which explains this simulation's stronger downdrafts. This occurred because updrafts were stronger in the lower-to-middle troposphere in the dry season owing to steeper low-level lapse rates and smaller entrainment rates, which resulted in stronger toroidal low pressure in rising cloud thermals.

Subsiding shell downdrafts play a crucial role in the transport of mass, heat, and momentum in the tropical atmosphere, all of which must be accurately assessed to maximize the accuracy of a climate model. Therefore, it is necessary that the driving mechanisms behind such shells are understood and applied to models. In addition to investigating the effects of environmental RH on shell behavior, it is recommended that steps be taken to properly implement subsiding shell and other near-cloud



downdraft effects into cumulus and convective parameterizations. To do this, future work points to investigating shells in a variety of environments, including those in a strongly-sheared environment. Leveraging observations of deep convective subsiding shell properties may also prove beneficial, such as those acquired during the Experiment of Sea-breeze Convection,
430   Aerosol, Precipitation, and Environment (ESCAPE) field campaign (Kollias et al., 2025). This, coupled with the continuation of LES and other numerical simulations, will lead to better comprehension of subsiding shells and encourage implementation of shell dynamics into climate models.

*Code and data availability.*   All model setup and analysis code is publicly available at https://figshare.com/account/home#/projects/262696.

*Author contributions.*   Experiment and model setup were constructed by QM and JP, with QM running the model simulations. All data
435   analysis code, except for the perturbation pressure decomposition code assembled by JP, was created and run by QM. QM prepared the manuscript with contributions from JP and JM.

*Competing interests.*   The authors declare that they have no conflict of interest.

*Disclaimer.*   Findings and conclusions of this research do not necessarily reflect the view of the United States Department of Energy or the National Science Foundation.

440   *Acknowledgements.*   This research was funded by United States Department of Energy grants DE-SC0022942 and DE-89243020SSC000051, as well as National Science Foundation grant AGS-2149353. Much appreciation is given to Jerry Harrington, Matt Kumjian, and Colin Zarzycki, all who provided feedback on this manuscript. Special thanks is given to Hugh Morrison at the National Center for Atmospheric Research for providing insight into this project and aiding in the setup of the simulations used for this research. Thanks is also given to Luke LeBel and Allen Mewhinney for their help with coding and analysis.



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
