# Peer review of "Driving Mechanisms for Subsiding Shells in Simulations of Deep Moist Convection"

_EGUsphere, 2025_

## Referee Comment (RC1)

**Review of submitted manuscript egu2025-4495 titled "Driving Mechanisms for Subsiding Shells in Simulations of Deep Moist Convection" by Mulhern et al.**

This study uses Large-Eddy Simulation (LES) experiments based on field campaign data in the Amazons to investigate drivers of subsiding shells surrounding deep convective clouds. The study builds on previous research, in particular findings on subsiding shells in non- or weakly-precipitating shallow convection. Two forces have previously been found to dominate the downward acceleration: i) negative buoyancy (B) due to evaporative cooling of cloud droplets, and ii) upward vertical pressure gradients (VPG) counteracting the downward acceleration. Debate still exists on the formation of convective downdrafts, in particular concerning the role of the vortex feature around the cloud and evaporative cooling at its edges. The main goal of this study is to gain more insight, and specifically to assess to what extent these shell-driving mechanisms differ in deep convective clouds. A key hypothesis to be tested is that drier deep cloud environments favor stronger downdrafts due to more efficient evaporative cooling, suggesting a dominant role by B in this regime. Another hypothesis tested is that differences in updraft strength contribute to differences in downdraft strength, through the VPG. These questions are then investigated using LES results and Lagrangian particle analysis.

The study and its findings are relevant not only for process understanding of convective downdrafts, but also for improving their representation in larger-scale models. The paper is well-written, in particular the introduction is informative and functions as a mini-review of past and recent research. The hypotheses to be tested are well-defined and relevant. The (numerical) experimental design and analysis method are in principle clearly described and well thought-through; however, a few potentially important aspects of the model setup are not as fully described or motivated as they could have been. This mainly concerns the spatial resolution (rather low I think) and the choice of microphysics scheme. I also have a few questions about the results of the simulations and their analysis, in particular i) the shape of the cloud field, the individual clouds, and the time evolution of the convective layer in general (currently not shown), and ii) the potential role of mixing of downdraft air with its environment in accelerating/decelerating the downdraft.

These main points are described in more detail below. In my opinion they can well be addressed by providing additional information and/or clarification.

**General recommendation:** Accept for publication after minor revisions.

**Main points:**

- i) The experimental setup of the LES makes sense, in that it contrasts dry versus moist conditions, reflecting seasonality in the Amazons. In that sense the study by Derbyshire et al. (2004, https://doi.org/10.1256/qj.03.130) comes to mind, which could be mentioned for reference (even though it did focus on bulk updraft mass flux, not downdrafts). However, I missed a few important details in the description of the model setup and the choices made:
  - Why is a spatial resolution of 100 m considered adequate for studying this problem? Most recent LES studies of moist convection typically use a finer gridspacing. Downdrafts can be pretty small and narrow, so a discussion of what resolution is adequate for capturing their dynamics would be beneficial for gaining confidence in the numerical results presented here. Ideally, a sensitivity test on resolution could be included (perhaps as an Appendix).
  - How is pressure treated in the model? There are various options out there on how to do this in an LES: prognostic, Poisson solver, etc. Line 160 briefly states that pressure is prognostic, but I am not entirely sure what this means. The treatment of pressure could affect how the VPG acts in downdrafts.
  - The microphysics scheme is minimally described, but could matter here, in particular through the evaporative cooling that partially drives downdraft acceleration. One question that comes up is if cloud ice plays a role in these simulations. The convective layer is tropical and pretty warm, but also very deep, and q\_i is mentioned in line 134.

- ii) I am missing a more detailed description of the general behavior of the simulation. This in particular concerns the cloud field. Do single clouds at all resemble the idealized picture of rising single cells as shown in Fig. 14? Do they consist of single rising thermals with a bipolar structure, or are they more complex, perhaps featuring more than one simultaneously? After all, one expects that some spatial organization takes place during these simulations. Such cloud shape heterogeneity could affect the downdraft behavior. The second aspect I was wondering about is the evolution during the time window of analysis. Is the convective layer really in steady state, or not really? Nudging is the only prescribed larger-scale forcing, and only applied above 4 km. This probably means that the atmospheric profile below that height is still evolving. It would be informative to show this, also because it affects the deep convection aloft.
- iii) Perhaps my moist important comment concerns the downdraft kinetic energy budget. It makes total sense to investigate downdraft acceleration due to B and VPG. But are these two forces considered to be the only contributors to the net downdraft acceleration? Judging from eq. (7) that seems to be the case. However, applying the common concept of an updraft model to a downdraft, I was thinking that diabatic mixing at the downdraft edge with its environment could also affect w. For example, a very recent study by Gu et al (2025, https://doi.org/10.1175/JAS-D-25-0017.1) also uses particle tracking, and shows that entrainment of air with different vertical velocity does affect the particle velocity budget, albeit for rising particles. I might be wrong, but I can imagine a similar mechanism also applying to sinking particles as part of a downdraft. I would recommend discussing the potential role of mixing on the net downdraft acceleration in Section 3.2, and perhaps also Section 3.4. A few more remarks on this:
  - In LES such mixing takes place through resolved advection and subgrid diffusion at downdraft grid points.
  - A quick check to find out if some term is missing could be to simply compare the sum of VPG and B to the net dw/dt (see Fig. 6). I assume the latter is diagnosed from the time series of w (Fig. 5)?
  - If downdraft dilution by mixing is important, does it have more impact in the dry season than the wet season? And a related question: does mixing between shell and updraft air accelerate or slow downdrafts?
- iv) My last point is more out of personal interest. Figures 5-9 show an impressive data collapse in downdraft properties as a function of time, both in w and its tendencies (B and VPG). But in Section 3.3 a significant variation in downdraft initiation height is shown (Fig. 10). Representing downdrafts in convection schemes requires information on downdraft initialization, so it is good to see it discussed here. Do downdrafts always initiate at the top of a convective cloud, as suggested in schematic Fig. 15? And if so, at what stage in the life cycle of the convective cloud? Or can downdrafts also start somewhere else, say, halfway the cloud? Plotting downdraft initialization height against convective cloud top height would give insight: is it a reasonable 1-to-1 relation? I understand that such analyses are not a direct objective of this study, so I leave it to the authors to include them or not.

**Detailed comments:**
* * *
Line 84: At this point the Derbyshire et al (2004) study could be mentioned. Even though it was not based on Amazonian conditions and used idealized profiles, it did focus on RH variation in the free troposphere, similar to what is observed in the Amazons.

Line 105: Why is 100m sufficient to resolve downdrafts in this convective regime? Would an anisotropic grid spacing, often used in LES, affect the results negatively?

Fig 1.: It would be informative to indicate the presence of the convective cloud layer in these diagrams, perhaps using grey background shading.

Line 122: The reader might not be aware what the default CM1 settings are. That includes the microphysics scheme, the choice of which probably affects the results. See also my  $2^{nd}$  main comment above. This would be a good point in the text to describe it.

Line 123: Was Newtonian nudging applied? I know such nudging is frequently applied in LES and CRM studies, perhaps referring to a few of those as a precedent would work well here. In particular to studies that focused on deep convection.

Line 130: An output time step of 30s is used. Do previous particle tracking studies suggest this time frequency is sufficient for resolving up/downdraft life cycles? I am thinking for example about the Hernandez-Deckers & Sherwood study (2016, DOI:10.1175/JAS-D-15-0385.1), what did they use to track thermals?

Line 160: Here prognostic pressure is mentioned. See my 1st main comment.

Line 196: The distance from the cloud edge of 800 m seems to be an important parameter in the analysis. Why was this value chosen?

Line 203: See my 2nd main comment, this would be a good point in the text to discuss the cloud field and the time evolution in general in a bit more detail

Fig 3: I always thought Hovmueller diagrams are plots of a horizontal cross section versus time. I am fully ok with calling these time-height plots Hovmuellers, but perhaps one could insert "Vertical" before.

Equation (6) and line 17: How dependent are the results on the choice of mixing length L? Where does this value come from, is it regime dependent (shallow vs deep)?

Fig. 5: One could plot w against normalized height as well. Does it show the same data collapse for both seasons? Is it worthwhile to add it as a  $2^{nd}$  panel here?

Fig 6c: How is net acceleration calculated, simply as the sum of B and VPG, or independently from the time change in w (Fig.5)?

Fig 10: Panel c is interesting, in that it seems to suggests that shells don't travel that far on average. Is there an analogy with the limited displacement distances found for rising thermals, in various recent studies (again the Hernandez-Deckers & Sherwood 2016 paper, and others)?